# Differences in the Expression Levels of SARS-CoV-2 Spike Protein in Cells Treated with mRNA-Based COVID-19 Vaccines: A Study on Vaccines from the Real World

**DOI:** 10.3390/vaccines11040879

**Published:** 2023-04-21

**Authors:** Luigi Cari, Mahdieh Naghavi Alhosseini, Antonella Mencacci, Graziella Migliorati, Giuseppe Nocentini

**Affiliations:** 1Section of Pharmacology, Department of Medicine and Surgery, University of Perugia, I-06129 Perugia, Italy; luigi.cari@unipg.it (L.C.);; 2Section of Microbiology and Clinical Microbiology, Department of Medicine and Surgery, University of Perugia, I-06129 Perugia, Italy

**Keywords:** mRNA-based vaccines, COVID-19, spike protein, real world

## Abstract

Comirnaty (BNT162b2) and Spikevax (mRNA-1273) COVID-19 vaccines encode a full-length SARS-CoV-2 Spike (S) protein. To evaluate whether the S-protein expressed following treatment with the two vaccines differs in the real-world context, two cell lines were treated for 24 h with two concentrations of each vaccine, and the expression of the S-protein was evaluated using flow cytometry and ELISA. Vaccines were obtained from three vaccination centers in Perugia (Italy) that provided us with residual vaccines present in vials after administration. Interestingly, the S-protein was detected not only on the cell membrane but also in the supernatant. The expression was dose-dependent only in Spikevax-treated cells. Furthermore, the S-protein expression levels in both cells and supernatant were much higher in Spikewax-than in Comirnaty-treated cells. Differences in S-protein expression levels following vaccine treatment may be attributed to variations in the efficacy of lipid nanoparticles, differences in mRNA translation rates and/or loss of some lipid nanoparticles’ properties and mRNA integrity during transport, storage, or dilution, and may contribute to explaining the slight differences in the efficacy and safety observed between the Comirnaty and Spikevax vaccines.

## 1. Introduction

The COVID-19 pandemic is the most serious medical emergency in contemporary history. With the joint effort of nations worldwide, effective vaccines were developed in a short time. Two mRNA-based vaccines, Comirnaty (BNT162b2) and Spikevax (mRNA-1273), have been shown to be considerably safer than adenovirus-based vaccines, especially regarding thrombotic adverse events with or without thrombocytopenia [1,2,3]. As a result, Comirnaty and Spikevax have become the primary vaccinations in use across Western countries, and their effectiveness in preventing severe COVID-19 and mortality is well known [4]. Despite the extremely rare adverse events in vaccine recipients, the benefits of vaccination vastly outweigh any risks. 

Comirnaty and Spikevax contain an mRNA strand enclosed in lipid nanoparticles (LNPs). These LNPs are quite different in composition and long-term stability at 4 °C [5,6,7]. The mRNA sequence encodes a full-length, stabilized SARS-CoV-2 Spike (S) protein, which comprises the extracellular receptor-binding domain and the COOH-terminal transmembrane domain. Despite coding the same protein, the mRNA sequence used in the Comirnaty and Spikevax vaccines differs [8].

Given the divergence between the two vaccines, we examined whether there was a difference in the levels of S-protein produced by the vaccines. Specifically, this study aimed to verify for the first time whether Comirnaty and Spikevax vaccines from a real-world context resulted in different in vitro levels of S-protein production.

## 2. Materials and Methods

Three vaccination centers in Perugia (Italy) provided us with residual vaccines present in vials after administration. The vials were collected between September and December 2021, during the first cycle of vaccination (first/second dose). The content of four vials of each vaccine was pooled under sterile conditions and used within 1 h after administration of the last dose. 

The non-adherent K562 and Jurkat human cell lines were purchased from ATCC (University Boulevard, Manassas, VA, USA). Cell lines were cultured in RPMI-1640 medium (Thermo Fisher Scientific, Waltham, MA, USA) containing 10% fetal bovine serum (Thermo Fisher Scientific, Waltham, MA, USA) and antibiotics (Thermo Fisher Scientific, Waltham, MA, USA). At 24 h before treatment with 1 and 10 µL of vaccine, cells exhibiting exponential growth were seeded on a 6-well plate at 0.25 × 10^6^ cells/mL (0.5 × 10^6^ cells in 2 mL). Untreated cells were used as the control. After 24 h of culture at 37 °C in a 5% CO_2_ atmosphere, cells were centrifuged at 300 rcf. To remove residual cell components, the culture supernatant was centrifuged again and stored at −80 °C. Cells were stained with a monoclonal antibody against the receptor binding domain of the S-protein (clone P05DHuRb; Thermo Fisher Scientific, Waltham, MA, USA) for 30 min. Cell surface expression of the S-protein was evaluated using flow cytometry (Attune™ NxT; Thermo Fisher Scientific, Waltham, MA, USA). Free S-protein in the cell culture supernatant was evaluated by ELISA in all the samples on the same day (Human SARS-CoV-2 RBD ELISA Kit; Thermo Fisher Scientific, Waltham, MA, USA). Four Comirnaty batches and two Spikevax batches were used for the measurement of cell surface S-protein expression, and four Comirnaty batches and four Spikevax batches were used for free S-protein measurement in the supernatant. Differences in S-protein expression were analyzed using the unpaired *t*-test (Prism 9.5.1; Graphpad software, San Diego, CA, USA).

## 3. Results

K562 and Jurkat human cell lines were treated with 1 and 10 µL of Comirnaty and Spikevax vaccines. We first assessed any potential cytotoxic effects. Appendix A show that the lower vaccine dose exerted nearly no cytotoxic effects on Jurkat cells and marginal cytotoxic effects on K562 cells. Moreover, Spikevax-treated samples displayed a higher proportion of dead cells compared with Comirnaty, particularly following the higher dose. 

The S-protein was detectable on the cell surface of both cell lines, as shown in Figure 1A–D, following treatment with both vaccines. Interestingly, a higher percentage of Jurkat cells exhibited cell surface S-protein expression compared with the K562 cells, as shown in Figure 1C,D.

Full-length S-protein, or a truncated form of it, was detected in the supernatant of both vaccine-treated cell lines (Figure 2A,B), with the K562 cell line showing much higher levels than the Jurkat cell line (Figure 2A,B). Thus, K562 cells exhibited lower cell surface S-protein expression levels, but higher levels of soluble S-protein than Jurkat cells, possibly suggesting that LNPs of vaccines favored the mRNA entry in both cell lines at similar levels and the observed differences in the detection of S-protein in the cell membrane and culture media reflect differences in the ability of the cells to release the S-protein in the supernatant. 

Notably, we found that Spikevax-induced S-protein expression levels were remarkably higher than those of Comirnaty, both on the cell membrane (Figure 1) and in culture media (Figure 2), in both cell lines.

## 4. Discussion

### 4.1. Differences in Antibody Response and Adverse Events between Comirnaty and Spikevax Vaccines

Our study demonstrates higher S-protein expression levels in cells treated with Spikevax compared with Comirnaty following an in vitro test. Interestingly, both vaccines showed detectable levels of the S-protein (or truncated forms of it) in the culture media. 

A vaccine’s effectiveness, including COVID-19 vaccines, depends on numerous variables, including the activity of the pandemic, vaccine recipients’ demographics, lifestyles, and ethnicities [9]. To compare the efficacy of different vaccines, the only way is to set up a randomized Phase 3 study comparing the same population receiving either the first or the second vaccine. However, no such studies have been performed. Comparative data from real-world studies suggest that Spikevax and Comirnaty vaccines have similar efficacy [4]. Interestingly, a study on 1647 vaccine recipients found that the anti-S-protein antibody titer after the second dose was approximately three times higher in Spikevax recipients than in Comirnaty recipients [10]. The higher mRNA dose in Spikevax (100 µg) compared to the Comirnaty dose (30 µg) potentially causes this difference. However, our results showing the different S-protein expressions by cell lines treated with Spikevax and Comirnaty vaccines may contribute to explaining the higher anti-S-protein antibody titer in Spikevax than in Comirnaty recipients. Notably, the differences in S-protein expression levels appear to be independent of the mRNA doses, because S-protein expression levels do not increase as a result of higher Comirnaty doses (Figure 1B), as discussed in Section 4.4.

Circulating S-protein has been shown to cause cardiovascular disease by damaging human heart pericytes through CD147 receptor binding and other unknown mechanism(s), regardless of viral infection [11,12] and myocarditis is the most common, though extremely rare, adverse event of mRNA-based vaccines [13]. Recent studies by Yonker et al. have shown that patients with myocarditis following COVID-19 mRNA vaccines had elevated levels of full-length S-protein, unbound by antibodies (free S-protein), in the plasma [14], agreeing with our findings that the S-protein is present in the supernatant of vaccine-treated cell lines. Interestingly, the full-length free S-protein was only present in vaccine recipients with myocarditis, strongly suggesting that the S-protein is responsible for vaccine-induced myocarditis [14,15]. 

Notably, myocarditis following vaccination with mRNA-based vaccines affects young males much more frequently than other demographics [13,16,17,18]. In these subjects, Spikevax shows a higher frequency of myocarditis than Comirnaty, with increased risk ranging from 2.5 to 8 folds in different studies [16,17,18]. The higher mRNA dose of Spikevax compared to Comirnaty is believed to be the reason for the increased incidence of myocarditis. If the different in vitro S-protein expressions by Spikevax and Comirnaty vaccines reflect in vivo conditions, our results could contribute to explaining the disparity in myocarditis frequency.

### 4.2. The Different Compositions of Comirnaty and Spikevax Vaccines May Account for the Varying Levels of S-Protein Expression

Our study did not explore the precise mechanisms underlying the different levels of S-protein expression induced by the Comirnaty and Spikevax vaccines. However, known information about the vaccines’ mRNA and LNP formulations provides some insights.

Both vaccines consist of a 5′-capped single-stranded mRNA encoding the full-length S-protein enclosed within LNPs. The S-protein produced by the two vaccines is identical and differs from the S-protein in the SARS-CoV-2 genome by two amino acids, which stabilize the resulting S-protein in the prefusion state to prime the host immune system to identify the virus before its entry into the host cell. In both vaccines, the mRNA is synthesized by in vitro transcription using a linear DNA template, and, as mammalian host cells attack unmodified exogenous RNA, all U nucleotides of the mRNA were substituted by N1-methylpseudouridine. 

Despite these similarities, the Spikevax and Comirnaty mRNAs differ in many other design features, such as the 5′- and 3′-untranslated regions and codon optimization [8]. Translation initiation, which is the rate-limiting step in translation, depends on the 5′-UTR, whereas if the translation initiation of an mRNA is highly efficient, translation elongation becomes rate-limiting and depends on codon optimization strategies. Therefore, differences in Spikevax and Comirnaty 5′-UTR and codons optimization strategies may be involved in regulating the synthesis rate of S-protein by mammalian cells. Notably, the S-protein mean fluorescence intensity (MFI) was only dose-dependent in Spikevax-treated cells (Figure 1B), possibly suggesting that mRNA features of Comirnaty may not allow rapid translation initiation and/or elongation becoming limiting for S-protein synthesis and independent of the levels of mRNA delivered into cells. However, more studies are needed.

The LNPs of Comirnaty and Spikevax are quite different [5,6]. Although both vaccines’ LNPs contain cholesterol and 1,2-distearoyl-snglycero-3-phosphocholine (DSPC), Comirnaty’s LNP contains ALC-0159 (a PEGylated lipid), whereas Spikevax’s LNP includes another PEGylated lipid (1,2-dimyristoyl-rac-glycero-3-methoxyPEG2000). Both contribute to nanoparticle stabilization [7]. A fourth LNP component is ALC-0315 [(4-hydroxybutyl)azanediyl)bis(hexane-6,1-diyl)bis(2-hexyldecanoate)] in Comirnaty’s LNP. ALC-0315 is an ionizable amino lipid responsible for mRNA compaction and aids mRNA cellular delivery [7]. The fourth LNP component is not mentioned by Spikevax manufacturer and, reasonably, differs from ALC-0315. Moreover, various vaccine excipients differ between Comirnaty and Spikevax. Therefore, differences in LNP composition and excipients may account for varying degrees of mRNA protection/integrity and mRNA delivery into cells.

### 4.3. The Differential Storage Conditions and Required Dilution of Comirnaty and Spikevax May Contribute to Explaining the Variations in S-Protein Expression Levels

It is plausible that the LNP composition and excipients play a role in dictating the different storage conditions for Comirnaty and Spikevax recommended by their respective manufacturers. The unpunctured vials of the latter can be stored at temperatures ranging from −50 °C to −15 °C until expiration, chilled and maintained between 2 °C and 8 °C for up to 30 days, and maintained at 8 °C to 25 °C for 24 h [19]. Meanwhile, unpunctured vials of Comirnaty can be stored at temperatures ranging from −90 °C to −60 °C until expiration, kept between 2 °C and 8 °C for up to 70 days, and held at 8 °C to 25 °C for 12 h [20]. The differential requirements of the Spikevax vaccine may suggest greater stability and efficacy in real-world settings, in line with our present study findings.

We observed that the Comirnaty vaccine required dilution prior to administration [21]. In contrast, there was no requirement for dilution of the Spikevax vaccine [22]. As Italy is a well-equipped country with well-trained healthcare personnel, we have no reason to conclude that the storage and possible dilution procedures of the vaccines in the tested samples were performed incorrectly. Nevertheless, this observation may hold implications for the LNP stability of Comirnaty in real-world contexts when compared with Spikevax.

### 4.4. The Differential Levels of In Vitro S-Protein Expression Do Not Depend on the Different mRNA Amounts in Comirnaty and Spikevax Vaccines

As previously stated in Section 4.1, Comirnaty and Spikevax vaccines utilized different mRNA doses during their administration. Specifically, when the ones that we tested were used (representing the first approved vaccines used in the first cycle of vaccination) adult subjects received 30 µg mRNA in 0.3 mL of Comirnaty and 100 µg mRNA in 0.5 mL of Spikevax. We supposed that vaccine manufacturers decided to use different mRNA doses to obtain similar levels of vaccine-induced S-protein. 

To verify this hypothesis, we conducted experiments utilizing 1 and 10 µL of both vaccines. In this way, we tested the effects of 0.1 and 1 µg mRNA present in the Comirnaty vaccine, and 0.2 and 2 µg mRNA present in the Spikevax vaccine. Therefore, when comparing the effects of 1 µL of Comirnaty and Spikevax, we compared the effect of a different dose of Comirnaty (0.1 µg of mRNA) and Spikevax (0.2 µg of mRNA), so that the different levels of S-protein observed on the cell membrane and culture media may be explained by the different mRNA dose. The same reasoning can be applied to the effects of 10 µL of Comirnaty and Spikevax.

However, our results also demonstrate that the varying levels of S-protein expression on the cell membrane and culture media are not solely attributable to differences in mRNA dose. Indeed, when comparing S-protein expression in cell lines treated with 1 µL Spikevax (0.2 µg mRNA) and 10 µL Comirnaty (1 µg mRNA), we observed a substantially lower level of S-protein expression in Comirnaty-treated cells (Figure 1C,D and Figure 2), despite the mRNA dose being five-fold higher than in Spikevax-treated cells. Consequently, it can be concluded that differences in S-protein levels following Spikevax and Comirnaty treatment are not due to differences in mRNA dose, but instead result from differences in mRNA sequence and/or LNP composition between the vaccines, as analyzed in Section 4.1. 

Notably, dose escalation of the Spikevax vaccine increases S-protein expression (Figure 1B (right panel) and Figure 2), whereas dose escalation of the Comirnaty vaccine does not (Figure 1B (left panel) and Figure 2).

### 4.5. Soluble S-Protein: Possible Explanations

The presence of S-proteins in the supernatant is unexpected given their hydrophobic -COOH terminal transmembrane domain that should not allow protein exit from the cells. Our observations, however, demonstrate the presence of S-protein (or truncated forms of it) in the supernatant, consistent with previous in vivo studies of vaccinated individuals [14,23,24]. 

Notably, K562 cells exhibited lower cell surface S-protein expression levels but higher levels of soluble S-protein than Jurkat cells, suggesting a varied ability of cells to release S-protein into the supernatant. Therefore, one possible hypothesis is that the S-protein can be shed by cells and the soluble S-protein represents the shed S-protein. 

Alternatively, the assessment report of the Spikevax vaccine by the European Medicine Agency (EMA) reports that “the applicant described short mRNA species that can occur because of abortive transcription or premature termination of transcription. As the majority of these short mRNAs do not contain a Poly A tail, the manufacturing process includes chromatography steps, which aim at removing these impurities to a large extent.” However, “additional bands are observed by an in vitro translation assay” performed using the purified mRNA [6]. A similar problem was reported by Comirnaty’s manufacturer, but, in the case of Comirnaty, the chromatography steps are not performed [5]. Therefore, it can be hypothesized that some of the short mRNA species present in the LNP of Spikevax and Comirnaty are translated into truncated S-proteins that do not have the transmembrane domain and can exit from the cells, explaining the in vitro finding of the present study and the in vivo finding from other authors [14,24].

If the rare adverse events of mRNA-based vaccines are due to the soluble S-protein, further studies are warranted to optimize the safety of mRNA-based vaccines and produce new vaccines even safer than the present ones.

### 4.6. Study Limitation

Further exploration into the underlying mechanisms responsible for the differential levels of S-protein expression observed with Spikevax and Comirnaty would be valuable, particularly to understand whether these differences are attributable to variations in LNPs, mRNA optimization, or both. However, our study did not test identical LNPs carrying the mRNA of either vaccine or identical mRNA delivered by LNPs present in either Spikevax or Comirnaty.

Additionally, it is important to note that the behavior of cell lines may not accurately reflect that of cells within the human body. In addition, our study utilized cell lines derived from the hematological compartment, with adherent cell lines intentionally omitted to prevent confounding biases due to detachment procedures. However, vaccines are injected into a muscle and their main target may be myocytes.

## 5. Conclusions

In conclusion, this study has compared the levels of S-protein expression in cells treated with SARS-CoV-2 mRNA-based vaccines used in the real world. Our results demonstrate that the Spikevax vaccine induces a higher level of S-protein expression in cells compared to the Comirnaty vaccine. These differences may be attributed to variations in LNP efficacy favoring mRNA entry in the cells, the mRNA sequence itself and/or loss of LNP and mRNA integrity during transport, storage, or dilution.

The observed findings have significant implications. Firstly, they may help explain the slight differences in the efficacy and safety of the Comirnaty and Spikevax vaccines. Secondly, they suggest the importance of dose–response studies, not only for new drugs but also for new vaccines. Lastly, our results might support the development of new-generation vaccines.

## Figures and Tables

**Figure 1 vaccines-11-00879-f001:**
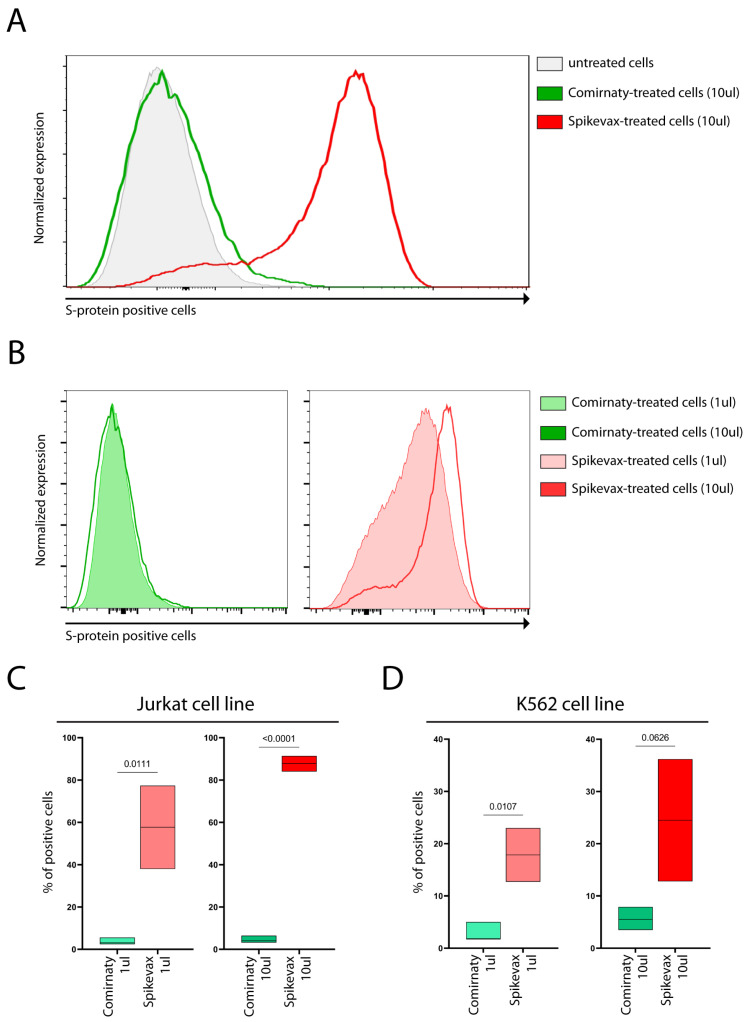
Flow cytometric SARS-CoV-2 Spike protein detection following treatment of cell lines with mRNA-based COVID-19 vaccines. Panel (**A**) shows the results of the flow cytometric evaluation of S-protein-positive cells in 10 µL Comirnaty-treated (green), 10 µL Spikevax-treated (red), and untreated (light gray) Jurkat cells (representative experiment). Panel (**B**) shows the flow cytometric evaluation of S-protein-positive cells in 1 µL (light green full area) and 10 µL (dark green) Comirnaty-treated Jurkat cells (left panel), as well as in 1 µL (light red full area) and 10 µL (dark red) Spikevax-treated Jurkat cells (right panel) (histograms generated from a representative experiment, the same of Panel (**A**)). Panel (**C**) (Jurkat cell line) and panel (**D**) (K562 cell line) show floating bars (min to max) depicting the percentage of S-protein-positive cells following 1 µL (left) and 10 µL (right) Comirnaty and Spikevax treatment; the line represents the mean value. The difference between Comirnaty and Spikevax treatment was evaluated using the unpaired *t*-test, and *p*-values are shown.

**Figure 2 vaccines-11-00879-f002:**
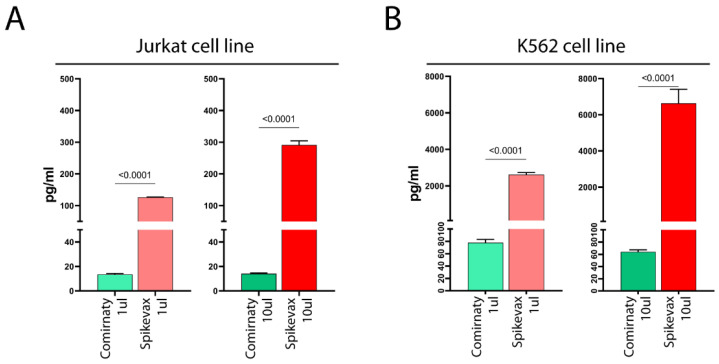
ELISA SARS-CoV-2 Spike protein detection following treatment of cell lines with mRNA-based COVID-19 vaccines. Panel (**A**) (Jurkat cell line) and panel (**B**) (K562 cell line) show the amount (pg/mL) of free S-protein in the supernatant of 1 µL (left) and 10 µL (right) Comirnaty-treated and Spikevax-treated cells, respectively. The mean ± SD is shown in the column bar graphs. Significant differences in S-protein detection between Comirnaty- and Spikevax-treated samples were assessed using the unpaired *t*-test and *p*-values are shown.

## Data Availability

The datasets used and/or analyzed during the current study are available from the corresponding author on reasonable request.

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
