# Peer review of "Differences in the Expression Levels of SARS-CoV-2 Spike Protein in Cells Treated with mRNA-Based COVID-19 Vaccines: A Study on Vaccines from the Real World"

_vaccines, 2023, doi:10.3390/vaccines11040879_

Round 1

Reviewer 1 Report

The authors have  compared two mRNA vaccines (Comirnaty (BNT162b2) and Spikevax (mRNA-1273)) in order  To evaluate whether the S-protein expressed following treatment  with the two vaccines differs in the real-world setting.  

Their  data appears to show that the Spikevax vaccine induces a  higher level of S-protein expression in cells than the Comirnaty vaccine. They suggest that this may explain why the frequency of myocarditis in Spikevax recipients is higher than that in Comirnaty re- 113 cipients.

This work suggests that the Spikevax vaccine should be utilized with caution going forward. The speed with which these vaccines were rolled out was in fact disturbing to some observers, because adequate tests such as described by these authors were not carried out.

I recommend the paper for publication.

Author Response

We thank the Reviewer for his/her positive comments on our manuscript.

Reviewer 2 Report

The manuscript by Cari et al "Differences in the expression levels of SARS-CoV-2 spike protein in cells treated with mRNA-based COVID-19 vaccines: A study on vaccines from the real world" presents a small, clearly structured study. Two cell lines were exposed to two concentrations of either BNT162b2 or mRNA-1273 vaccine for 24h and the expression of spike protein was studied. Importantly, the S-protein was detected not only on the cell membrane but also released in the supernatant. Expression was dose-dependent, and a stronger expression on cell surface and release into supernatant was found for mRNA1273 compared to BNT162b2. These findings are interesting in relation to both, vaccination efficacy and side effects for the two mRNA vaccines. In particular the potential correlation of released S-protein and the side- effects, such as myocarditis are addressed in the discussion. 

There are two points of critizism:

1. the choice of the cell lines. Both cell lines used represent suspension cell lines, both of hematopoietic origin, with only limited relevance for S-protein expression in vivo after i.m. mRNA delivery. Instead of chosing two hematopoietic cell lines, testing also in a cell type  to be expected to primarily express the S-protein after mRNA delivery in vivo  would have added to the study

2. the authors do not provide mechanistic data for explanation of the observed significantly stronger S-proetin expression and release for mRNA1273 compared to Comirnaty, showing only that the difference can NOT be explaned by the different mRNA doses in the two vaccines.

While differences in the LNP efficacy and mRNA integrity are discussed, no data in this line are provided, which otherwise could have enriched the study

The authors are encouraged to add such data if possible.

Generally, in the context of cited literature the study provides new interesting pieces of data for solving the puzzle of the observed side-effects of mRNA vaccination, such as post-vac myocarditis

Author Response

We thank the Reviewer for the helpful suggestion provided; please find attached the Point-by-point Response to the Reviewer’s comments.

Reviewer 3 Report

Thank you for allowing me to review the article titled ‘Differences in the expression levels of SARS-CoV-2 spike protein in cells treated with mRNA-based COVID-19 vaccines: A study on vaccines from the real world’The main objective of this study was to investigate, for the first time in a real-world scenario, whether there is a difference in the levels of S-protein produced by the two vaccines. This study provides important guidance for individuals planning to receive COVID-19 vaccination in choosing which mRNA vaccine to receive. his study requires an evaluation of the immunogenicity of the expressed S protein, not just an evaluation based solely on the expression level. In my opinion, this paper should be published after minor revision.

Author Response

We thank the Reviewer for his/her positive comments on our manuscript; please find attached the Point-by-point Response to the Reviewer’s comments.

Round 2

Reviewer 2 Report

The authors addressed the points raised by the reviewer in an appropriate way, providing an optimized revised manuscript.

The article can be recommended for publication in Vaccines